# Parents’ Knowledge of and Attitude toward Acute Otitis Media and Its Treatment in Children: A Survey at Primary Healthcare Centers in the Aseer Region

**DOI:** 10.3390/children10091573

**Published:** 2023-09-19

**Authors:** Hayfa Abdulkhaleq AlHefdhi, Maraam Mohammed Al Qout, Alhanouf Yahya Alqahtani, Meshal Mohammed Alqahtani, Roaa Mohammed Asiri, Omair Mohammed Alshahrani, Hanan Delem Almoghamer, Naglaa Youssef, Ramy Mohamed Ghazy

**Affiliations:** 1Department of Family and Community Medicine, College of Medicine, King Khalid University, Abha 61421, Saudi Arabia; dr.hhefdhi@gmail.com; 2College of Medicine, King Khalid University, Abha 61421, Saudi Arabia; maraammalqout@gmail.com (M.M.A.Q.); alhanouf.q98@gmail.com (A.Y.A.); mshal8888mm@gmail.com (M.M.A.); roaamohammad.2001as@gmail.com (R.M.A.); dr.omair2@gmail.com (O.M.A.); 9ihdm9@gmail.com (H.D.A.); 3Department of Medical-Surgical Nursing, College of Nursing, Princess Nourah Bint Abdulrah-Man University, Riyadh 11671, Saudi Arabia; nfyoussef@pnu.edu.sa; 4Tropical Health Department, Alexandria University, Alexandria 21561, Egypt

**Keywords:** Aseer region, Saudi Arabia, otitis media, antibiotic resistance, health literacy, health education

## Abstract

Background: Acute otitis media (AOM) in children aged 5 years old and younger poses a critical health concern, affecting both the general health of children and the emotional well-being of parents. The objective of this study was to evaluate parental understanding, attitudes, and experiences related to AOM and its management, including the use of antibiotics without physician prescription. Method: A cross-sectional study was carried out during the months of May and June of 2023. To collect data for this study, a validated questionnaire was converted into a Google form and given to parents of children aged 5 years and younger who sought healthcare for their children in primary healthcare centers in Abha city, Aseer region, Saudi Arabia. Results: A total of 406 parents participated in this study, 64.8% of them were women and 45.3% of them were in the age range of 25 to 34 years. The majority (90.1%) resided in urban areas and a significant proportion (72.2%) had a university education. Among the respondents, 22.7% agreed that bacteria are the cause of AOM, while 21.7% agreed that it is caused by viruses. A total of 51.0% acknowledged the need for antibiotics in the management of AOM. In terms of treatment, 84.5% and 83.5% believed that analgesics and antibiotics, respectively, were the most effective for otalgia. A substantial portion, 43.1%, 34.7%, and 37.7%, respectively, believed that antibiotics could reduce pain, relieve fever, and prevent recurrence. Almost three-quarters sought medical advice primarily from paediatricians and 37.7% obtained information about AOM from the Internet. Furthermore, almost a third (28.8%) chose not to wait for the physician’s appointment and immediately administered antibiotics without the physician’s prescription due to concerns about disease progression. Approximately two fifths (38.4%) requested physicians to prescribe antibiotics, a pattern that was in agreement with the actual rate of antibiotic prescriptions (38.4%). Conclusions: A notable deficiency in knowledge and unsafe practices about AOM and its management is evident among parents in the Aseer region. This underscores the pressing need for an educational program aimed at improving parental health literacy regarding otitis media causes and treatments, as well as its preventive measures.

## 1. Introduction

Otitis media (OM) is a group of illnesses that include acute otitis media (AOM), otitis media with effusion (OME; ‘glue ear’), and chronic suppurative otitis media (CSOM) [1]. OM is a significant contributor to healthcare visits, which is considered a preventable cause of hearing loss, particularly in developing countries [2]. OM in children aged 5 years and younger poses a critical health problem in developing countries, affecting both the general health of children and the emotional well-being of parents. AOM is an infectious disease seen mainly in children [3]. Annually, AOM affects 11% of the population, while CSOM affects 5% of the population, with children under 5 years of age accounting for 50% and 22.6% of these cases, respectively. At the age of three years, a substantial portion, ranging from 50% to 85%, of children experience at least one episode of AOM [2]. Children aged 6 to 24 months develop AOM more frequently. Furthermore, the risk decreases significantly after age 5 years [3]. OME is characterized by the presence of middle ear effusion (MEE) behind an intact tympanic membrane; however, in contrast to AOM, OME is not associated with signs and symptoms of an acute infection [4]. The main symptom of OME is a conductive hearing loss caused by impaired transduction of sound waves in the middle ear due to the presence of MEE [1]. CSOM is defined as chronic inflammation of the middle ear and mastoid cavity; persistent or recurrent ear discharge through a tympanic membrane perforation or a ventilation tube is the most prominent symptom. CSOM causes conductive hearing loss and might damage the middle ear ossicles. It also increases the risk for permanent sensorineural hearing loss (hearing loss due to damage to the inner ear) and intracranial complications [5].

There are two types of risk factors for acquiring AOM: host and environmental. The first group includes age, gender, ethnicity, family history of AOM and genetic predisposition, craniofacial anomalies, atopy, immunodeficiency, adenoid hypertrophy, and gastroesophageal reflux [1]; environmental factors include daycare attendance, passive smoking, older siblings, use of a pacifier, not breastfeeding [1], pollution, season, and delivery route [6]. It is worth noting that during the coronavirus disease (COVID-19) pandemic, school closures and physical distance measures were related with a decrease in the prevalence of AOM and favored the resolution of chronic forms among children who attended the outpatient clinic [7].

Common symptoms of AOM include otalgia (which could be noted by actions such as rubbing, tugging, or holding the ear), fever, irritability, discharge from the ear, bulging of the tympanic membrane, loss of appetite, and occasionally vomiting or a state of lethargy [3]. AOM can lead to complications such as persistent perforation of the tympanic membrane, hearing loss, and, in some cases, severe outcomes such as neck abscesses, mastoiditis, meningitis, and labyrinthitis [8]. Hearing impairment resulting from AOM can delay language development, hinder academic performance, and impact future employment opportunities. The chronic form often goes unnoticed and is frequently misdiagnosed due to its lack of pain [9].

In young children, AOM is one of the main reasons for prescribing antibiotics [10]. The choice of antibiotic use requires careful consideration of the advantages and disadvantages they offer, as their prescription contributes to the emergence of antibiotic resistance. Antibiotic resistance is a growing issue in public health, affecting not only the broader population but also individual patients, who can experience bacterial resistance to a specific antibiotic for up to 12 months after its use [11]. Although existing guidelines advise against antibiotics in the majority of cases of children with AOM, a significant number continue to receive antibiotic treatment [12].

Previous research indicates that various sociodemographic factors among parents, such as their educational background, age, and the number of children they have, can impact their knowledge and attitudes about AOM and its treatment [13]. Studies have shown that the use of shared decision making (SDM) during medical consultations is greatly influenced by the level of health literacy among parents. Limited health literacy tends to foster a hierarchical relationship between physicians and parents, increasing the likelihood of following physician recommendations [14]. Consequently, supporting and enhancing the interaction between parents and physicians, along with fostering a better understanding between these two parties, emerges as a promising strategy for promoting rational antibiotic prescription for children affected by AOM.

In Saudi Arabia, the incidence of AOM was found to be more frequent in children up to 4 years of age, while it was less common among those aged 8 to 12 years of age. Among the sexes, male children exhibited a slightly higher AOM rate compared to female children. The prevalence of AOM among children varied according to their geographical location, with higher rates in the southern and central regions compared to other provinces [15]. Alsuhaibani et al. [13] reported that a significant part of the population in Qassim city, Saudi Arabia, had suboptimal knowledge about AOM, where 56% showed positive attitudes towards it, and 86.6% did not perceive vaccination as an effective preventive measure against AOM. Similarly, Al-Hammar et al. [16] reported that 85.39% of participants in the Al-Ahsa province exhibited an inadequate level of knowledge about AOM.

There is a notable lack of research investigating parental awareness and behaviours concerning AOM in Saudi Arabia, particularly with the use of a validated questionnaire. Our hypothesis postulates that there exists a correlation between parental knowledge levels and their attitudes toward AOM, particularly the pattern of antibiotic usage among parents. This study aims to evaluate parental understanding, attitudes, and experiences related to AOM and its management, including the use of antibiotics without physician prescription.

## 2. Materials and Methods

### 2.1. Study Design and Study Setting

This cross-sectional study was carried out during the months of May and June of 2023. To collect data for this study, a validated questionnaire was uploaded to a Google form and distributed to parents of children aged 5 years and younger who sought healthcare for their children in primary healthcare centers (PHCs) in the Aseer region, Saudi Arabia. The total area of Aseer is 81.000 km^2^, and the population is estimated at 1,563,000 people. It is situated in the southwestern area of Saudi Arabia and stands as one of the country’s administrative regions, with the emirate’s headquarters located in Abha. Parents were personally approached in 11 PHCs in Abha city.

### 2.2. Study Population and Sample Size

Using the convenience sampling technique, we recruited Saudi parents who had at least one child aged 5 years or under and who were seeking medical care for their children in primary healthcare centers. We included parents aged 18 years or older, who had smart phones or computers with access to the Internet. Using Epi-Info 7.2, the minimum sample size was calculated to achieve a 95% confidence level with a margin of error of ±0.05, assuming that the population proportion of having good knowledge is 0.50. This yielded a minimum required sample size of 384. We increased the sample by 5% to compensate for non-response.

### 2.3. Study Outcome

The primary objective of this questionnaire was to evaluate and quantify parental understanding, attitudes, and personal experiences concerning AOM and its management. Our assessment encompassed various aspects, including their knowledge of AOM’s causes, the role of different medications (such as antibiotics, analgesics, and herbals), their perceived effectiveness, as well as their perspectives on the disease’s progression. We delved into their preferred contact person when dealing with a case of AOM and identified their primary source of information about the condition. Furthermore, we explored their practices regarding the initiation of antibiotic treatment. Lastly, we examined the frequency of medication use for treating AOM and the medications they recommend physicians to prescribe for their children.

### 2.4. Study Questionnaires and Data Collection

The data were collected using a two-part questionnaire.

Part 1 collected data about the sociodemographic characteristic of the parents including their gender, age, residence, and education level. Additionally, it collected data about characteristics and health conditions of the children, which included data about the type of health insurance, the child’s age, nursery attendance, and pacifier use.

Part 2 collected data about parents’ knowledge of and experience with AOM.

This questionnaire was originally developed by Sibylle et al., 2015 [17], to measure parental understanding, attitudes, and experiences related to AOM. This questionnaire had three sections: The first section collected data on the causes and impact of ear inflammation, treatment of pain associated with ear inflammation, and the effectiveness of antibiotics in the treatment of ear inflammation. The second section aimed to explore caregivers’ preferences, sources of information, and attitudes towards the treatment of AOM in their children; identify which individuals’ opinions matter most to caregivers when it comes to treating AOM; caregivers’ preferences for sources of information about AOM; and attitude towards “wait for physician’s appointment and treatment strategy with antibiotics’’. The third section included frequency of AOM episodes, which healthcare provider they consulted most frequently in cases of AOM, how often they asked a doctor to prescribe certain drugs for their child’s AOM (medicine with pain-relieving/fever-reducing substances, antibiotics, naturopathic with remedies, ear drops with pain-relieving substances, and nasal drops with a decongestant), and how often their doctor prescribed the same set of drugs for their child with AOM. They could choose responses ranging from “always” to “never” for each drug category. Appendix A.

### 2.5. Questionnaire Validation

To establish the questionnaire’s validity, an initial step encompassed a forward translation of the questionnaire into formal Arabic, carried out independently by two bilingual coauthors. Each item and its corresponding response options were evaluated by the coauthors in terms of translation difficulty. The questionnaire was then back-translated into English. Minor inconsistencies were identified and resolved through collaborative discussions between researchers. The validity of the content was evaluated through the input of an expert panel comprising four members. This panel consisted of methodologists, otolaryngologists, public health experts, and language specialists. The experts critically reviewed the translated version to ascertain whether it adequately encompassed the intended concepts as defined. Subsequently, we proceeded with cognitive testing of the pre-final version. Specially trained members of the research team conducted cognitive interviews with 20 participants in the target respondent group. These interviews were designed to assess various aspects including participants’ comprehension, readability, linguistic accuracy, phrasing, cultural relevance of items, and the clarity of response instructions within each section.

### 2.6. Ethical Apporval and Consent

The study objectives were effectively communicated to parents and their informed consent was acquired prior to the start of the study. The ethical approval for this study was granted by the Research Ethics Board of the University of King Khalid (ECM # 2023–2015) in adherence with the ethical principles outlined in the Declaration of Helsinki. Parents were duly notified that they had the option to withdraw their participation from the study at any point, without impacting their child treatment process or relation with physician. Additionally, they were informed that the information collected is confidential and restricted in access, limited solely to the principal investigator and the statistician. It was explicitly conveyed that these data would not be utilised for any other publication.

### 2.7. Statistical Analysis

Statistical analysis was carried out using the R software version 4.1.1. Categorical variables were expressed as frequencies and percentages. Cronbach’s alpha coefficients were computed to evaluate the internal consistency of the questionnaire. As a general guideline, a Cronbach alpha value falling within the range of 0.70 to 0.80 is regarded as satisfactory for research purposes, whereas an alpha exceeding 0.80 is considered indicative of a very strong level of internal consistency [18]. To compare two independent categorical variables, Pearson’s chi-square test was used. A *p*-value below 0.05 was considered statistically significant for this analysis.

## 3. Results

### 3.1. Sociodemographic Characteristics of Parents and Children

Table 1 presents the distribution of the variables studied based on parental and child criteria in a sample of 406 participants. In terms of parental criteria, the gender distribution shows that 263 participants (64.8%) were females, 45.3% were between 25 and 34 years old, and 90.1% resided in urban areas. In terms of education, 72.2% had a university education. When switching to child-related criteria, the health insurance distribution shows that 50.7% of the children were covered by governmental insurance. The age distribution of the children indicates that 13.3% were under one year old, 14.8% were one year old, 19.7% were four years old, 12.8% were two years old, 18.0% were three years old, and 21.4% were five years old. Furthermore, 86.5% of the children did not attend nursery and 83.7% of the children did not use a pacifier.

### 3.2. Parental Knowledge of Otitis Media

In relation to the statement “Caused by bacteria”, 22.7% fully agreed. Similarly, for the statement “Caused by viruses”, 1.7% of participants strongly disagreed, while 21.7% of them fully agreed. The opinions of the parents on the statement “Associated with intensive earache” showed that 30.8% fully agreed. On the idea that AOM “resolve spontaneously”, responses varied from 6.7% strongly disagreeing to 14.5% fully agreeing. A notable trend is observed in the statement “Needs antibiotic treatment”, with only 0.7% strongly disagreeing, while 51.0% fully agree that it requires antibiotic treatment (Figure 1a).

Regarding the best treatment for otalgia, the majority (84.5%) fully agreed that a pain reliever/fever reducing substances is effective, while 3.4% disagreed entirely. For ‘antibiotics’, a high proportion (83.5%) fully agreed with their efficacy to reduce pain while only 1.5% completely disagreed. In the case of “Naturopathic remedies”, 23.3% agreed completely on their value and 17.5% disagreed entirely. Regarding ear drops with a pain reliever substance, a significant 90.8% fully agreed on their effectiveness. For “Nasal drops with decongestant”, 58.3% fully agreed and 8.3% did not agree at all. Lastly, “Household remedies” garnered mixed responses, but smaller proportions fully agreed (16.5%) (Figure 1b).

Concerning the opinion of the respondents on the role of antibiotics in the treatment of AOM, 43.1% fully agreed that antibiotics lead to rapid pain relief, 34.7% fully agreed that antibiotics lead to rapid reduction of fever in children, 37.7% fully agreed that antibiotics generally reduce the likelihood of relapse of AOM, 36.5% fully agreed that antibiotics generally reduce the risk of permanent ear damage, 17.2% fully agreed that antibiotics negatively affect the stomach and intestine of children, 21.7% fully agreed that antibiotics negatively affect children’s immunity, and 33.3% fully agreed that antibiotics may become ineffective after frequent use (Figure 1c).

### 3.3. Parental Attitudes towards Otitis Media

When participants were asked about the contact partners that are the most important to you in the case of an AOM, the fully agreeing responses were as follow; general practitioner (22.9%), paediatrician (70.9%), close relatives (16.5%), parents of other children (21.3%), teachers in child care facilities (10.3%), and friends who are health care professionals (22.4%) (Figure 2a) When asked about the most important source of information sources on AOM in children, 15.8% fully agreed that the most important sources were newspapers and magazines, 19.5% fully agreed on books, 16.5% fully agreed on radio and television, and 37.7% agreed on the Internet (Figure 2b).

Regarding the wait-and-see strategy, for the statement “I am willing to wait and only use antibiotics when symptoms persist two days”, 42.6% fully agreed. Regarding the statement “I am willing to wait and only use antibiotics when symptoms do not improve or even worsen overnight”, 34.2% fully agreed. For the statement “I am willing to wait. In the case of persistent symptoms, I consult the doctor again before using an antibiotic”, 47.8% fully agreed. For “I am not willing to wait and use antibiotics when my child severely suffers from symptoms”, 31.5% fully agreed. Finally, in the statement “I am not willing to wait and use antibiotics immediately because I am concerned that the disease might get worse”, 28.8% fully agreed (Figure 2c).

### 3.4. Parent’s Experience with Acute Otitis Media

Regarding the parent’s experience with antibiotic prescription, 57.0% of parents always ask the physician to prescribe analgesics; 38.4% always ask them to prescribe antibiotics; 14% ask for neuropathic remedies; 30.2% ask for ear drops; and 29.1% ask for decongestant (Figure 3a). The physician always prescribes analgesics in 52.3% of cases, antibiotics in 38.4%, neuropathic remedies in 12.8%, ear drops in 32.6%, and nasal decongestant in 26.7% (Figure 3b).

Among the various sociodemographic factors and parental knowledge and attitudes examined, the factors most strongly linked to the incidence or recurrence of AOM were parents’ belief in the detrimental impact of antibiotics on their children’s immunity, and parents’ immediate preference for antibiotic use due to concerns about the potential worsening of the illness (Table 2).

In this study, 86 (21.2%) children experienced AOM and nearly 5.8% (5/86) had more than 10 attacks. Figure 4 provides an overview of parents’ attitudes and behaviors related to seeking medical advice for AOM. It highlights the association between parents’ requests for physicians to prescribe analgesics and the recurrence of AOM. Among the responses, it is notable that 73% of parents whose children experienced 3–10 episodes of AOM consistently requested physicians to prescribe analgesics. Additionally, 60.0% of parents whose children had more than 10 AOM episodes also made such requests. Finally, 56.5% of parents whose children had experienced less than 3 AOM asked physicians for analgesic prescriptions. A statistically significant difference was observed in the responses, as indicated by the *p*-value of 0.007.

## 4. Discussion

Our research focused on assessing parental comprehension, attitudes, and experiences related to AOM and its management, particularly focusing on the pattern of antibiotic use. Almost a fourth of the participants identified bacteria and viruses as the main aetiology of the disease, while more than half fully agreed that AOM requires antibiotics for treatment. Only 14.5% considered that AOM is a self-limited disease and can resolve spontaneously. More than three-quarters of the participants fully agreed on the role of ear drops, antibiotics, and analgesics in reducing otalgia. Almost half of the parents acknowledged the role of antibiotics in reducing pain, while about a third thought that it reduces fever, prevents relapse of AOM, and prevents permanent ear damage. Parents often turn to paediatricians for medical advice when their child is affected by AOM, while the Internet was the main source of information. Approximately one-third of the parents fully agreed to use antibiotics immediately without delay. In terms of their experiences, 21.2% of children experienced AOM; nearly 5.8% had more than 10 attacks. Approximately two-fifths of parents consistently asked physicians to prescribe antibiotics, and a similar percentage noted that healthcare providers invariably prescribe antibiotics for the management of AOM.

This study emphasized a prevalent misconception about the effectiveness of antibiotics in the treatment of AOM, in contrast to a more pragmatic understanding of their potential adverse outcomes. Drawing on their knowledge, parents saw antibiotics as essential for addressing AOM, with a strong belief in their ability to alleviate pain and reduce symptom recurrence. In fact, prompt relief of pain is of a significant importance to parents due to the substantial burdens that AOM places on affected children and their families [19]. On the other hand, Venekamp et al. [20] reported that the anticipated rapid analgesic response (within 24 h) of antibiotics as part of AOM treatment in children was not confirmed. Antibiotics do not cause substantial pain reduction in the first 24 h. Furthermore, antibiotics do not effectively reduce the incidence of severe complications or the recurrence rate of AOM. Interestingly, a substantial proportion of the respondents indicated their willingness to take antibiotics preemptively, even before symptoms emerged. In the bivariate analysis, this practice was significantly associated with the incidence of AOM. Interestingly, most of the participants showed limited awareness of the potential side effects and complications associated with antibiotic use. This lack of recognition was reflected in the significant proportion of respondents who regularly requested physicians prescribe antibiotics. This observation aligns with the findings reported by other researchers [21,22]. On the other hand, strong parental recognition of the increased risk of antimicrobial resistance associated with excessive antibiotic usage, as evidenced in this study, was reported in many other studies [21,23].

Most of the respondents agreed that AOM causes intense ear pain. That is why a major sector of the studied population believed that analgesics are the best treatment for otalgia. Interestingly, our findings revealed a notable connection between parental suggestions for prescribing analgesics and the recurrence of AOM. We theorise that the effects of pain relief medications might lead parents to halt or even abandon the treatment regimen. There are uncertainties surrounding the origins and natural progression of the disease. A small sector of parents, around 22.7%, hold a general agreement that bacteria are responsible for causing AOM, while 20.7% generally attribute AOM to viruses. In particular, a relatively small proportion of parents admit to being unaware of the cause, with figures of 10.6% for bacteria and 10.3% for viruses. Similar findings were reported by Sibylle et al. [17] in Germany. Therefore, the dominant understanding that viruses predominantly contribute to the pathophysiological mechanisms of AOM appears to lack widespread recognition [24,25]. Parents who have witnessed cases of AOM themselves tend to be significantly less inclined to believe in the involvement of viruses, compared to parents without direct experience with AOM. This observation is particularly significant considering the general strong awareness of antibiotics’ efficacy against bacterial infections [21,26], as shown in our findings. Merely, 14.5% of the participants in this survey generally concur with the concept that AOM typically resolves on its own. These points of view are reinforced from the perspectives of more than half of parents who generally agreed that AOM requires antibiotic treatment. These figures indicate that parents may significantly underestimate the inherent tendency of simple AOM in children to resolve without intervention. This belief is further validated by a previous survey that reported a notable proportion of parents who believe antibiotics are indispensable for the treatment of AOM (85% in Finland and 55% in The Netherlands) [27].

International guidelines, along with more recent national assessments, advise that children presenting with uncomplicated unilateral AOM should initially undergo observation coupled with symptomatic treatment, using analgesic drugs as the primary approach. Antibiotics should be considered if symptoms do not improve in 48 to 72 h. The use of alternative agents, such as naturopathic remedies, ear drops with pain relief properties, or nasal drops with decongestants, is not explicitly endorsed [25]. In this study, there is a very high rate of antibiotic prescription. There is a correlation between the rate of actual prescription of medications and parental suggestions to prescribe specific drugs, including antibiotics. This finding challenges the prevalent idea that parents often pressure physicians to prescribe antibiotics for their children with AOM [28]. Our analysis reveals that the frequency of experiencing AOM does not substantially affect parental prescription of medications, except for analgesics. This suggests that a general familiarity with children’s health might exert a more substantial influence on parental knowledge and attitudes toward AOM and its treatment than specific experience with AOM itself.

A significant discovery from this study is that almost half of the participants actively seek information about AOM on the Internet. This phenomenon presents a dual-edged sword, as it exposes them to the risk of encountering misinformation that could influence the progression of the disease. A systematic review conducted by Suarez-Lledo et al. [29] reported that the prevalence of health misinformation was more prominent in studies related to products such as smoking and substances, including opioids and marijuana. In some studies, the dissemination of misleading information through posts reached as high as 87%. The misinformation on vaccines was also notably widespread (43%), with the human papillomavirus vaccine bearing the brunt. However, these online channels offer convenient means of reaching a wide segment of the population and providing accurate information to them. It has been shown to be very effective in controlling and managing different health conditions [30], including the coronavirus disease of 2019 (COVID-19) [31].

The outcomes of this study emphasize the important need for focused educational programs and healthcare policies aimed at improving parental understanding and responsible management of AOM. It is critical to prioritize culturally sensitive and region-specific approaches to improving health literacy. Future study should explore deeper into intervention effectiveness, taking into account the impact of healthcare practitioner supervision, the function of online information sources, and children’s long-term health outcomes. Furthermore, investigating strategies to reduce antibiotic overuse and gaining a better understanding of the factors influencing parental decision-making regarding AOM management will be critical in promoting safer and more informed practices, ultimately improving the overall health and well-being of children and their families.

### Strengths and Limitations

Our study presents a notable strength as it is a pioneering survey focusing on the knowledge, attitudes, and experiences of Saudi parents with children aged 5 years and younger, specifically with regard to AOM and its treatment, with a particular emphasis on antibiotics use. Additionally, we use a validated questionnaire to ensure the internal validity of the study findings. Although not striving for complete representativeness, given that we focused solely on the Southern region, a region known for its high incidence of AOM, the results still offer preliminary insights into parental viewpoints on AOM and its management. However, specific limitations apply to this survey. The sample of participants is relatively small, and parental recruitment is based on a convenience sampling method. There is a potential non-response bias, considering only parents who voluntarily participated, potentially skewing the sample. Recall bias might also be present due to the time gap between experiencing the condition and providing responses from memory. Additionally, the inherent limitation of a cross-sectional survey restricts our ability to establish causality.

## 5. Conclusions

A significant portion of the surveyed population had inadequate knowledge of AOM, particularly its causative agents and the potential for spontaneous resolution. A considerable number of participants believed that antibiotics were effective in alleviating pain, complications, and recurrence of AOM. In particular, we identified substantial rates of antibiotic prescriptions that correlated with parental suggestions for antibiotic use. Around a third of the participants were fully in favour of immediate antibiotic use. Based on the research results, it is recommended to launch focused educational initiatives targeting parents, to enrich their understanding of AOM, its treatment possibilities, and the importance of antibiotics, while highlighting the inherent self-restraint of AOM. Forging collaborations with healthcare professionals to improve communication and adherence to evidence-based protocols, increasing parents’ proficiency in digital health literacy, and cultivating a collaborative parent-paediatric relationship emerge as crucial approaches.

## Figures and Tables

**Figure 1 children-10-01573-f001:**
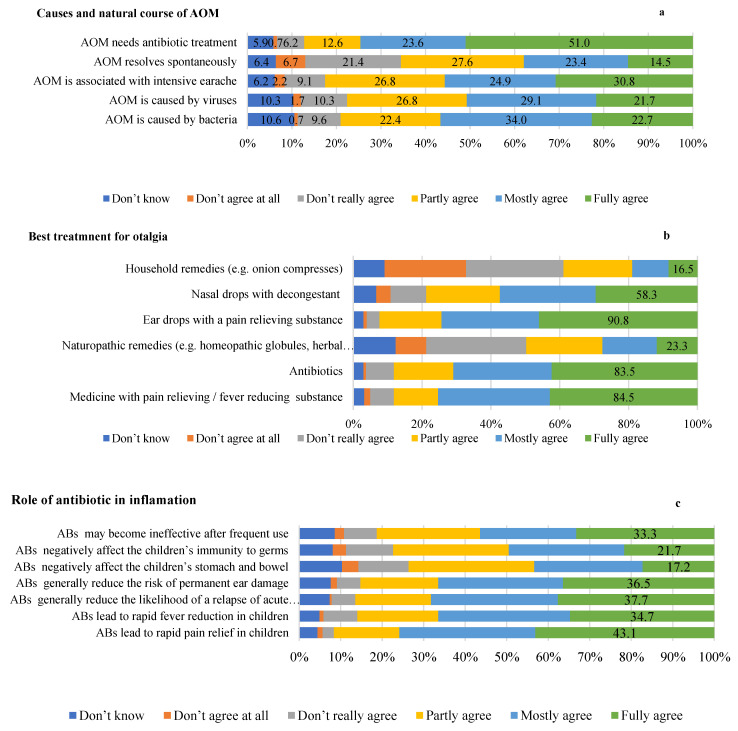
Parents knowledge about acute otitis media; (**a**): knowledge of parents regarding causes and natural course of AOM, (**b**): parents’ knowledge about best treatment regimen of otalgia, (**c**): Role of antibiotic in treating middle ear inflammation.

**Figure 2 children-10-01573-f002:**
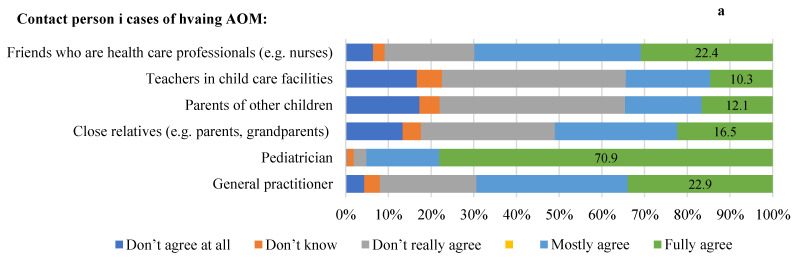
Parents attitude towards management of acute otitis media.

**Figure 3 children-10-01573-f003:**
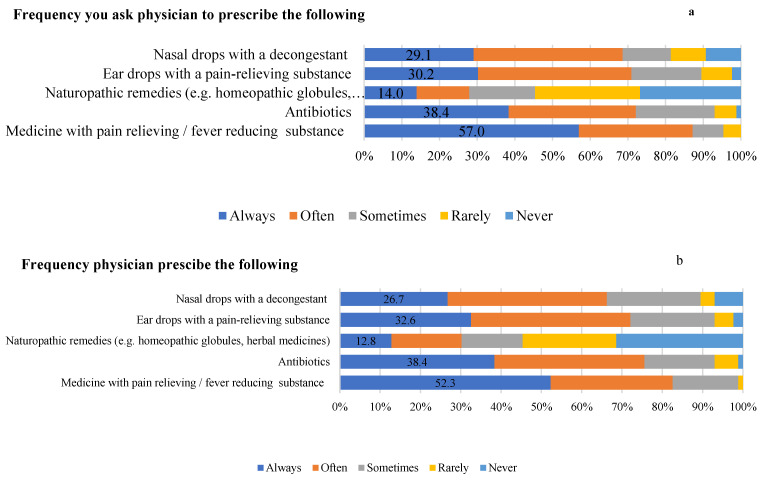
Parents practice regarding management of acute otitis media.

**Figure 4 children-10-01573-f004:**
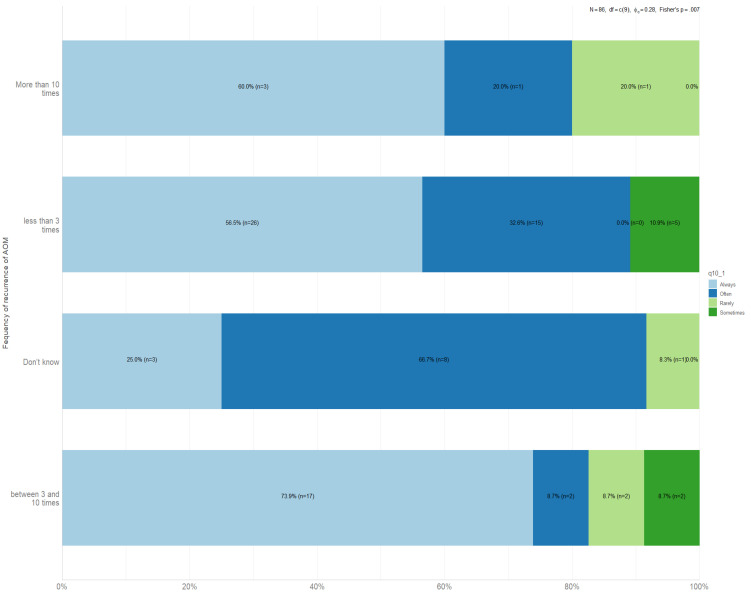
Association between recurrence of AOM attacks and parental recommendations for physician-prescribed analgesics.

**Table 1 children-10-01573-t001:** Demographic Distribution of Parents and Children (N = 406).

N = 406	Variables		
Parents criteria	Gender	Female	263 (64.8%)
		Male	143 (35.2%)
	Age	From 18 to 24 years	26 (6.4%)
		From 25 to 34 years	184 (45.3%)
		From 35 to 44 years	162 (39.9%)
		From 45 to 54 years	27 (6.7%)
		From 55 to 64 years	7 (1.7%)
	Residence	Rural	40 (9.9%)
		Urban	366 (90.1%)
	Education	Below university	113 (27.8%)
		University	293 (72.2%)
Child criteria	Health insurance	Governmental	206 (50.7%)
		Private	43 (10.6%)
		None	157 (38.7%)
	Age child	Below one year	54 (13.3%)
		One year	60 (14.8%)
		Two years	52 (12.8%)
		Three years	73 (18.0%)
		Four years	80 (19.7%)
		Five years	87 (21.4%)
	Attend nursery		55(13.5%)
	Use a pacifiers		66(13.5%)

**Table 2 children-10-01573-t002:** Factors associated with the incidence of acute otitis media.

Dependent:		Did Not Experience AOM	Experienced AOM	*p*
Antibiotics negatively affect children’s immunity to germs.	Don’t know	28 (8.8)	5 (5.8)	0.029
	Don’t agree at all	13 (4.1)	0 (0.0)	
	Don’t really agree	31 (9.7)	15 (17.4)	
	Partly agree	95 (29.7)	18 (20.9)	
	Mostly agree	90 (28.1)	23 (26.7)	
	Fully agree	63 (19.7)	25 (29.1)	
I am not willing to wait and use antibiotics immediately because I am concerned that the disease could get worse.	Don’t know	16 (5.0)	0 (0.0)	0.008
	Don’t agree at all	21 (6.6)	7 (8.1)	
	Don’t really agree	55 (17.2)	9 (10.5)	
	Partly agree	65 (20.3)	29 (33.7)	
	Mostly agree	75 (23.4)	12 (14.0)	
	Fully agree	88 (27.5)	29 (33.7)	

## Data Availability

Data are available upon request by emailing the corresponding author.

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
