# Peer review of "Parents’ Knowledge of and Attitude toward Acute Otitis Media and Its Treatment in Children: A Survey at Primary Healthcare Centers in the Aseer Region"

_children, 2023, doi:10.3390/children10091573_

Round 1

Reviewer 1 Report

I would like to thank the authors for their submission and allowing me to review their work.

This is an interesting study on an important topic. However, I would be grateful if you could add further explanations and changes on the following points:

1) INTRODUCTION: Page 2, line 55

In the introduction section, I would highlight the key role of upper respiratory tract infections (URTIs) and social interaction in the pathogenesis of otitis media in young children (I suggest citing the following article: Effects of COVID-19 Lockdown on Otitis Media With Effusion in Children: Future Therapeutic Implications. Otolaryngol Head Neck Surg. 2021;165(5):710-715. doi:10.1177/0194599820987458)

2) INTRODUCTION: Page 2, line 59

I suggest adding “bulging of the tympanic membrane” as a typical sign of acute otitis media. I would clarify the differences between otitis media, otitis media with effusion (OME), acute otitis media (AOM), and chronic otitis media (COM).

(References: Otitis media. Nat Rev Dis Primers. 2016;2(1):16063. doi:10.1038/nrdp.2016.63).

3) RESULTS: Page 4, line 179

To make the percentages clearer to read, I would also put the exact proportion in brackets.

4) RESULTS: Page 4, line 186

I suggest specifying the mean age (± standard deviation) and gender of the children.

5) TABLE 1: Page 5, line 191

I think it would be interesting to specify whether the parents had one or more children. Having other older children could influence the answers to the questionnaire.

6) DISCUSSION: Page 12, line 365

Which are the future prospects of this study?

Good

Author Response

I would like to thank the authors for their submission and allowing me to review their work. This is an interesting study on an important topic. However, I would be grateful if you could add further explanations and changes on the following points:

Thank you for taking the time to review our submission. We genuinely appreciate your efforts and valuable feedback. Your input plays a crucial role in improving the quality of our work. We are pleased to hear that you found our submission worthy of appreciation, and we are grateful for your kind words. Your encouragement motivates us to continue our research and contribute to the field.

1) INTRODUCTION: Page 2, line 55

In the introduction section, I would highlight the key role of upper respiratory tract infections (URTIs) and social interaction in the pathogenesis of otitis media in young children (I suggest citing the following article: Effects of COVID-19 Lockdown on Otitis Media with Effusion in Children: Future Therapeutic Implications. Otolaryngol Head Neck Surg. 2021;165(5):710-715. doi:10.1177/0194599820987458)

Response: We have cited this article.

2) INTRODUCTION: Page 2, line 59

I suggest adding “bulging of the tympanic membrane” as a typical sign of acute otitis media. I would clarify the differences between otitis media, otitis media with effusion (OME), acute otitis media (AOM), and chronic otitis media (COM).

(References: Otitis media. Nat Rev Dis Primers. 2016;2(1):16063. doi:10.1038/nrdp.2016.63)

Response: We have added this sign and cited this article.

3) RESULTS: Page 4, line 179

To make the percentages clearer to read, I would also put the exact proportion in brackets.

Response: done.

4) RESULTS: Page 4, line 186

I suggest specifying the mean age (± standard deviation) and gender of the children

Response: Sorry for inconvenience. we collected data regarding child age in categorical form not continuous quantitative variable.

5) TABLE 1: Page 5, line 191

I think it would be interesting to specify whether the parents had one or more children. Having other older children could influence the answers to the questionnaire.

Sorry for inconvenience. we did not collect this variable.

6) DISCUSSION: Page 12, line 365

Which are the future prospects of this study?

We added the following paragraph

The outcomes of the study emphasize the important need for focused educational programs and healthcare policies aimed at improving parental understanding and responsible management of AOM. It is critical to prioritize culturally sensitive and region-specific approaches to improving health literacy. Future study should explore deeper into intervention effectiveness, taking into account the impact of healthcare practitioner supervision, the function of online information sources, and children's long-term health outcomes. Furthermore, investigating strategies to reduce antibiotic overuse and gaining a better understanding of the factors influencing parental decision-making regarding AOM management will be critical in promoting safer and more informed practices, ultimately improving the overall health and well-being of children and their families.

Many thanks for this great revision

Reviewer 2 Report

The authors wrote an interesting and well-documented article. Although the subject has been previously explored in studies from different countries, it could have significant relevance for the region that the authors have specifically chosen to focus on. However, there are some areas for improvement and suggestions that I will detail in the next paragraphs:

1. The results section might be difficult to follow. I recommend merging the "mostly agree" category with the "fully agree" category and "don't agree at all" with "don't really agree" to enhance the clarity and readability of the findings. This will reduce the number of variables and will make the results more concise.

2. I recommend adjusting the titles of the figures to more accurately depict their contents. Some titles may be inadequate, such as "wait and see" in Figure 2c, which could be revised to "the parents' willingness to delay antibiotic treatment."

3. The questionnaire used for the study should be included as supplementary material.

4. I recommend including more recent references, preferably published in the last 5 years in order to increase the accuracy of the statements

5. It is unclear why the authors only detail the association between the recurrence of attacks and the recommendation of prescribing analgesics and not other associations. It would be helpful to provide more explanations for this. Additionally, the figure lacks a proper legend and the text showing the percentages is partially unreadable.

6. In the Methodology section, within the study outcome subsection, it would be beneficial to provide a more detailed enumeration of the specific objectives related to "parental understanding, attitudes, and personal experiences related to AOM and its management." This will enhance readability and clarity in the subsequent results section.

The text should be revised to correct any grammatical errors (for example, "ethical approval" instead of "ethical apporval").

Author Response

The authors wrote an interesting and well-documented article. Although the subject has been previously explored in studies from different countries, it could have significant relevance for the region that the authors have specifically chosen to focus on. However, there are some areas for improvement and suggestions that I will detail in the next paragraphs:

Thank you for your thoughtful and constructive feedback on our article. We appreciate your positive remarks regarding the article's interest and documentation. We recognize the importance of building upon existing research, and your point about the regional relevance is well taken.

The results section might be difficult to follow. I recommend merging the "mostly agree" category with the "fully agree" category and "don't agree at all" with "don't really agree" to enhance the clarity and readability of the findings. This will reduce the number of variables and will make the results more concise.

We completely agree with your comment. However, we have adhered to the statistical analysis plan established by the authors who designed the questionnaire. If you believe it is necessary, we can consider consolidating the responses.

  1. I recommend adjusting the titles of the figures to more accurately depict their contents. Some titles may be inadequate, such as "wait and see" in Figure 2c, which could be revised to "the parents' willingness to delay antibiotic treatment."

Required change Done

  1. The questionnaire used for the study should be included as supplementary material.

We uploaded the study questionnaire.

  1. I recommend including more recent references, preferably published in the last 5 years in order to increase the accuracy of the statements

We tried to update references as possible and removed some old references.

  1. It is unclear why the authors only detail the association between the recurrence of attacks and the recommendation of prescribing analgesics and no other associations. It would be helpful to provide more explanations for this. Additionally, the figure lacks a proper legend and the text showing the percentages is partially unreadable.

I apologize for any confusion. Your focus on the prescription of analgesics makes sense, especially since it was the only significant factor associated with the recurrence of AOM. Thank you for clarifying your approach. We have added a new figure with better resolution. Sorry for inconvenience.

  1. In the Methodology section, within the study outcome subsection, it would be beneficial to provide a more detailed enumeration of the specific objectives related to "parental understanding, attitudes, and personal experiences related to AOM and its management." This will enhance readability and clarity in the subsequent results section.

Thanks, we added more details in this section.

Thank you very much for such a great review

Round 2

Reviewer 1 Report

The authors have clarified my questions.

Reviewer 2 Report

I appreciate your prompt response to my requirements. Thank you for your efficiency.